# Changes in the Well-Being of Foreign Language Speaking Migrant Mothers Living in Finland during the Initial Stage of the COVID-19 Pandemic

Eveliina Heino [1,*], Hanna Kara [2] and Camilla Nordberg [2]

1  Faculty of Social Sciences, University of Helsinki, 00014 Helsinki, Finland
2  Department of Education and Welfare Studies, Åbo Akademi University, 65100 Vaasa, Finland;
   hanna.kara@abo.fi (H.K.); camilla.nordberg@abo.fi (C.N.)
*  Correspondence: eveliina.heino@helsinki.fi

**Abstract:** This article examines changes in the well-being of foreign-language-speaking migrant mothers living in Finland during the initial stage of the COVID-19 pandemic in the spring of 2020. Our data consist of 73 mothers' responses to a qualitative survey conducted between 18 April and 26 May 2020. In our analysis, we employ the division of well-being into three dimensions: having, loving, and being. According to our results, the participating mothers experienced dramatic changes, such as an increased burden of care and domestic work, difficulties helping children with remote studies, health concerns, a lack of free time, isolation from Finnish society and the inability to travel to their country of origin. Family-centered activities helped the mothers to cope in this situation but also caused strains. Based on our findings, we discuss the vulnerabilities these mothers experienced in relation to language, migration background and gender roles.

**Keywords:** COVID-19; foreign language speaking mothers; migrants; well-being; Finland

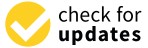



## 1. Introduction

In January 2020, the World Health Organization declared COVID-19 a public health emergency of international concern (World Health Organization 2020), and in mid-March, the Government of Finland declared a state of emergency (The Emergency Power Act 1552/2011 2011). After this declaration, education was suspended in schools and other institutions. Most kindergartens were open, but the Finnish government recommended that children under school age should be cared for at home. Public sector employees whose jobs allowed them to work remotely were assigned to remote work and employees in other sectors were recommended to work remotely (Finnish Government 2020a, 2020b). Leisure activities for families with children were also suspended or restricted and social services substantially decreased their services or offered only remote services, which reduced social support for families (Hastrup et al. 2021).

Studies show that the COVID-19 pandemic influenced the everyday lives of families with children in many ways. Childcare or supporting children's schooling along with work and housework proved to be a challenge for many families. This led to parental fatigue and was reflected negatively in family relationships (Salin et al. 2020; Singletary et al. 2022). Economic concerns due to layoffs and many services closing caused economic stress for many households (Paju 2020; Lee et al. 2020).

Despite this, some parents also experienced positive changes in their everyday lives as, with no commuting, hobbies or scheduled busyness, families could spend more time together (Salin et al. 2020; Sorkkila and Aunola 2021; Toran et al. 2021). The pandemic seemingly caused a particular strain on families who already experienced stress and were in a vulnerable position, thus deepening social inequalities (Helske et al. 2021).

Studies have shown a higher risk of infection in migrant populations compared to the general population, as well as difficulties in accessing health services and economic resources, a loss of income, and increased social isolation and discrimination (Adhikari et al. 2022; Guadagno 2020; Nisanci et al. 2020; Rafieifar et al. 2021; Shaaban et al. 2020). The pandemic reinforced the existing mobility regimes and negatively influenced the transnational practices of migrants due to lockdowns and travel restrictions between countries (Kempny 2022).

However, there is a lack of peer-reviewed studies regarding foreign-speaking mothers or/and migrant mothers during the COVID-19 pandemic. This is noteworthy because the underrepresentation of a particular group in research can be regarded as reflective of their perspective also lacking in the discussion and planning of policies and services, further exacerbating this. This study aims to address this gap in research and explore the experiences of mothers with a first language other than one of the national languages, i.e., Finnish or Swedish, who migrated to Finland and who lived in Finland at the beginning of the COVID-19 pandemic. We ask: *what changes did the COVID-19 pandemic bring for foreign-language-speaking migrant mothers' well-being*?

In this study, we use the concept *migrant* while referring to a heterogeneous group of mothers who have moved to Finland from other countries. We acknowledge that migration can be a different experience for individuals, e.g., because of different reasons for migration. However, migrancy often involves various structural vulnerabilities related to a renegotiation of social status positions in the destination country (Nordberg 2020).

The factors creating structural vulnerability include, for example, legal precarity, because migration often entails obtaining temporary legal status, and not always knowing about its continuation. Secondly, migrants are at a high risk of encountering racism in the host country (European Union Agency for Fundamental Rights 2023). It is also noteworthy that the public discussion and public policies concerning migration are heavily politicized in many countries, which can affect the ways in which migrants perceive their own position and rights in the host society (Heino 2018). Fourthly, migration affects families in many ways, limiting daily face-to face interaction between family members living in different countries and the provision of physical social support (see Kara and Wrede 2022). Sometimes unpredictable border regulations and changes in the international relations between nation-states can have huge effects on transnational relations (Hiitola et al. 2020; Nordberg et al. 2024). Fifthly, speaking the majority language/s is often a prerequisite to access and make use of public services, to enter the job market and for daily interaction with the majority population (Heino et al. 2023a; Scheibelhofer et al. 2020).

Previous pre-pandemic and non-COVID-19-related studies (e.g., Nyikavaranda et al. 2023) also indicate that migrant women are, in general, at a higher risk of suffering from mental-health-related problems and distress compared to male migrants and the majority population. Risk factors for female migrant populations include social isolation, discrimination, and financial hardship (also Aroian et al. 2008).

In this study, we approach the social category of motherhood as a construct with varying connotations across different contexts. That is, motherhood, on the one hand, is influenced by norms, institutions, cultural expectations, and power relations, while on the other hand, constantly constructed by mothers within the context of everyday care (Berg 2008; Nordberg 2015, 2020). We agree with Hillier and Greig (2020) in that studying mothers' experiences, especially in the early stages of the pandemic, is important; this is because, during that time, additional unpaid caregiving, which has usually been expected of women, increased. From a social perspective, it is important to create a better understanding of the influence that the COVID-19 pandemic had on the everyday lives of families with children. These effects can now be seen in social work practice, for example, in an increased number of service users and more severe problems faced by children and parents, including decreased mental health (Calcaterra and Landi 2023; Heino et al. 2023b).

Our data consist of 73 mothers' responses to a qualitative survey distributed through social media between 18 April and 26 May 2020 in five languages. The respondents

represented 30 countries. We employ a theoretical framework based on Erik Allardt's (1976) three dimensions of well-being: *having* (standard of living and physical needs), *loving* (social relationships), and *being* (self-actualisation). Although Allardt originally presented the theory of well-being in the 1970s, it is still relevant today as it addresses well-being as a multidimensional phenomenon consisting of material, social and psychological dimensions (Uusitalo and Simpura 2020).

Our study is qualitative, and our focus lies on subjective well-being as the respondents in our data defined their own well-being and highlighted the factors that they perceived relevant. The COVID-19 pandemic was a new situation for everyone and no extensive research data regarding its impact on individuals' well-being were available at that time. Qualitative research on well-being is particularly important during societal change as it creates space for respondents' own definitions, which can highlight issues that researchers have not acknowledged as relevant (Teti et al. 2020).

## 2. Migrants' Well-Being during COVID-19

Migration and transnational mobility remain topical issues in Finland. By the end of 2022, more than 477,000 migrants were living in Finland, representing 9% of the current population. The number of migrants has grown rather rapidly, as in 1990 the number of migrants was 64,922 (Statistics Finland 2023).

Many different aspects, such as age, the cause of migration, country of origin and time lived in the host country, affect migrants' well-being. Furthermore, definitions of well-being vary across studies and different contexts. In a broad study on the well-being of migrants implemented in Finland, the subjective well-being of the participants was lower than for the majority population. Experiences of discrimination also hampered the participants' sense of security. Both physical and mental health risks were shown to be higher within migrant groups that have a refugee background, related partly to the trauma background (Kuusio et al. 2020).

Earlier studies have suggested that COVID-19 had somewhat different consequences for migrant and majority populations. Firstly, the different effects were health related, as in Finland, especially in the Helsinki capital area, the relative share of foreign language speakers among those infected has been higher than that of Finnish speakers (Holmberg et al. 2022). This has been explained, for example, by a lack of adequate multilingual and targeted information and cramped living conditions, as well as the concentration of migrant workers in service professions where remote working was limited (Skogberg et al. 2021; Backholm and Nordberg 2023). The high proportion of migrants among those infected has also been noted in other OECD countries (OECD 2020).

In addition, differences have been found when comparing the amount and strength of migrants' concerns with the concerns of the majority population during COVID-19. The migrant population reported higher levels of concern in relation to, for example, getting infected with COVID-19 themselves, infecting others, a close one becoming infected, being discriminated because of having infection, the continuation of employment, the government's ability to deal with the crisis, the healthcare system's ability to treat all patients, and loneliness (Skogberg et al. 2021). Other international studies with similar results indicate a decrease in migrants' psychological well-being (Garrido et al. 2023).

Attention has also been paid to the labour market position of foreign language speakers during the pandemic, and both in Finland and internationally, unemployment among foreign language speakers and migrants has increased relatively more than unemployment among the majority population (OECD 2020).

## 3. Mothers' Well-Being during COVID-19

According to previous studies, gender inequalities increased during the COVID-19 pandemic. A large survey aiming to measure the impact of COVID-19 on women's lives was conducted in the EU countries and showed that, for 38% of the respondents, the pandemic had a negative impact on their income, and for 44%, it influenced their work–life balance

(Eurobarometer 2022). Studies from the US (Collins et al. 2021), Canada (Hillier and Greig 2020), Italy (Manzo and Minello 2020), Germany (Zoch et al. 2021), the Netherlands (Yerkes et al. 2020), Australia (Craig and Churchill 2021), Hungary (Fodor et al. 2021), France (Chatot et al. 2023) and Finland (Lahtel and Pietiläinen 2022) have shown that women, and especially mothers, experienced an increased burden of care and domestic tasks during the pandemic, which affected their well-being in many ways. Czymara et al. (2021) showed that the COVID-19 pandemic influenced women more than men because women perform more physical care tasks, such as housework and homeschooling, and organize childcare; meanwhile, men feel more responsible for paid work. Wandschneider et al. (2022) and Thorsteinsen et al. (2022) found that working mothers especially experienced guilt because the norms of the 'ideal mother' and the 'ideal worker' were in conflict, worsening their mental health.

Different studies employ various concepts and focus on different aspects of mothers' well-being, but all of them indicate declines in mental health. Wandschneider et al. (2022) found that a lack of support with childcare was associated with higher levels of depressive symptoms among mothers. Möhring et al. (2021) showed that, especially for mothers, the lockdown had a negative effect on satisfaction with family and work. According to Cameron et al. (2020), maternal depression and anxiety appeared to increase in the context of the COVID-19 pandemic compared to previously reported results.

Most current studies focus on the effects on mental health, and it is possible that effects in other areas, such as physical health, will be recognized later. It is also noteworthy that mothers do not form a homogenous group and that their social positions, such as ethnic origin, social class and family structure, shape their everyday lives in various ways. For example, mothers who cared for children with disabilities faced many difficulties early in the pandemic due to disruptions in the education and rehabilitation processes of their children and increased social isolation (Çelik 2023). Combining work and family life was difficult for single mothers because they could not share the additional burden or have the flexibility to combine work and domestic duties (Hertz et al. 2020).

The diversity of the effects of the COVID-19 pandemic across different social categories has been covered more in North American research. In their study, Bastain et al. (2022) found that, while more changes in everyday life caused by the pandemic and higher PTS were reported by socioeconomically advantaged respondents, the association between PTS and higher pandemic-associated hardships, coping mechanisms, and behavioral changes was greater in the less socioeconomically advantaged cluster. In a Canadian study, Hirani and Wagner (2022) showed that refugee mothers were at a high risk of experiencing add-on stressors due to isolation, difficulty in accessing health care, COVID-19-related restrictions in hospitals, limited follow-up care, limited social support, financial difficulties, and compromised nutrition. Another Canadian study (Lim et al. 2022) highlighted migrant mothers' experiences of social isolation and loneliness during the pandemic due to their status as migrants, which led to a lack of social support and negatively affected their mental well-being.

## 4. Data Description and Analysis

The article forms part of a larger research project "SOSKIELI (Linguistic diversity and vulnerability in social work in the era of digitalisation)", funded by Ministry of Social Affairs and Health, and implemented at Åbo Akademi University and University of Helsinki. The analysis draws from a qualitative survey targeted towards parents with young children during the initial stage of the COVID-19 pandemic in spring 2020. The survey was distributed on social media with the help of social work organizations, Facebook groups for families with children, blogs, as well as the Åbo Akademi University and University of Helsinki communications departments; the survey was available in Finnish, Swedish, Russian, English and Spanish. These languages were selected based on the language skills of the research team, as there was no external funding available for translations at that time. Consequently, some languages of large migrant populations in Finland, such as Estonian,

Arabic and Somali, were not covered. However, Russian is the most widely spoken foreign language in Finland, English is the most widely used foreign language as a lingua franca, and Spanish was able to reach respondents from both Europe and Latin America.

The survey consisted of nine open-ended questions covering issues such as how everyday life and care were organized in families with children at the time and how respondents experienced the new situation (see Appendix A for the list of the open-ended questions). We also asked for the following background information: age (roughly), gender, educational level, number of minor children and adults living in the household, age of children (roughly), region of residence and country of birth.

The survey was conducted from 17 April to 30 June 2020 and received over 1500 responses. The number of foreign language responses, i.e., responses given in languages other than Finnish or Swedish, was 95. From these answers, 73 respondents identified themselves as women who were born outside of Finland and their responses form the data in this article. The respondents were divided by languages as follows: English, 32 respondents; Russian, 33 respondents; and Spanish, 8 respondents. The responses were received between 18 April and 26 May 2020, either during the state of emergency or shortly after its official end on 27 April 2020.

Most respondents were aged between 30 and 45. Over half (65%) had a university degree, and most (69%) lived in the Helsinki capital region. A little under half (46%) of the respondents lived with one child, and 35% with two children, most of whom were kindergarten (56%) or primary school (47%) age. Roughly a third (34%) of the respondents lived in a single-parent family.

The study adheres to the ethical research guidelines ascribed by the Finnish National Board on Research Integrity TENK (TENK 2019). Information regarding the study (aims of the study, data production, handling and storage, publication of the results) and contact information of the researchers was available in the information letter and at the end of the questionnaire in all five languages. In the extracts used in this article, all details that could identify the participants have been omitted. We will also not specify the original languages in the extracts to protect participant anonymity.

Our primary research question for this article is: *what changes did the COVID-19 pandemic bring for foreign-language-speaking migrant mothers' well-being*? We answer this question by analyzing the qualitative responses to nine open-ended questions concerning possible changes in families' everyday lives. The total length of the data analyzed for this article is 50 pages.

In our analysis, we have employed the framework of well-being developed by Erik Allardt (1976) to guide our interpretations. Allard's study is considered a staple in well-being research in Finland and beyond, and continues to be discussed and employed (see for example Kattilakoski and Sireni 2020; Tuukkanen and Pekkarinen 2023). Allardt's work was seminal in operationalizing well-being in terms of the conditions necessary for human development and existence, paving the way for a holistic understanding of families' situations and social needs, rather than seeing problems mainly as constituted by parental neglect and shortcomings (Engen et al. 2021). According to Allardt (1976), well-being is a state in which it is possible for human beings to satisfy their needs. It consists of three dimensions: *having* refers to material conditions, such as standards of living and physical needs including housing, working conditions, and health. *Loving* covers social interaction with family as well as communities. *Being* includes possibilities of self-realization, such as personal growth, leisure-time activities and political participation.

Our analysis draws from a thematic analysis (Braun and Clarke 2022). We aimed to create a dialogue between our observations from the empirical data and the well-being theory of Allardt. The first author created an initial coding chart which included the classification of the data according to the three dimensions of well-being. Next, the first and second authors examined and worked on this division, and tackled the similarities and differences they recognized, creating a revised coding chart aimed at more explicitly addressing and naming the themes within each well-being dimension. In the final stage, all

three authors reflected and elaborated each theme within the well-being framework. At this stage, we especially focused on the changes the participants described in their well-being, since change was central in the responses. We also elaborated factors that were related especially to language and migration within each theme.

We identified and named the following themes visible in Table 1:

**Table 1.** Results of analysis.

| Dimension of Well-Being | Themes | Factors Related to Language and Migration within Themes |
| --- | --- | --- |
| Having | - Worsening of daily living conditions<br>- Health concerns<br>- Increase in care and domestic work | - Lack of social contacts and support in Finland<br>- Uncertainty regarding information about national public health instructions and restrictions<br>- Difficulties with assisting children with studying remotely |
| Loving | - Tensions in relationships with other household members<br>- Disrupted face-to-face relationships with family living abroad<br>- Weakening ties with the Finnish society | - Homesickness, cultural mourning, and worries related to the pandemic's development and care responsibilities abroad<br>- Everyday life without social connections to Finnish society<br>- Decreasing in Finnish language skills |
| Being | - Lack of own and free time<br>- Increase in family-centered activities | |

Under the dimension *having*, the respondents described *a worsening of their daily living conditions* as well as *an increase in health concerns* and *care and domestic tasks*. Under the dimension *loving*, the respondents outlined changes in their relationships such as *tensions in their relationships with other household members*, *disrupted face-to-face relationships with family living abroad* and *weakening ties with Finnish society*. Under the dimension *being*, the respondents reported negative changes, such as a *lack of own and free time*, but also an *increase in family-centered activities*, which helped mothers to cope in the new circumstances.

In the following sections, we will discuss our analytical findings in further detail. We employ quotes from the data to illustrate our interpretations.

## 5. Having

### 5.1. Worsening of Daily Living Conditions

The respondents described changes in their everyday living conditions during the initial stage of the pandemic as radical because all family members were at home in the same space. Parents started remote work, school-aged children started remote studies and under school-aged children stayed at home. Activities were cancelled and public spaces where families spent free time before the pandemic, such as libraries and indoor playgrounds, were closed. Sharing the same space with all family members caused different kinds of changes to mothers' daily lives. The most important change was decreasing private space:

> *It is hard to be so close to each other all the time in the apartment. Nowhere to 'let off steam' and hard to have time to think or focus on something as there is always someone around.* (17)

When changes in daily living conditions were described as negative in the responses, this was often in connection to not having the physical space to be alone and being constantly interrupted. In addition, constant noise in the apartment hampered concentration and caused irritation and stress. In larger families, mothers were also responsible for

resolving arguments between children. Many respondents wrote how they understand why their children were restless and frustrated, but that their own exhaustion causes them to lose their patience which then led to feelings of guilt:

> *I must keep them from fighting with each other because of spending 24 h together, they are frustrated and want their own space, a change of scenery, a return to normalcy. I often snap at them or tell them to be quiet and leave the room—they are not particularly noisy, but I am overwhelmed.* (4)

Mothers explained that feeling overwhelmed affected interactions with their children. "Snapping", "irritated speech", and "shouting" caused guilt. Maternal guilt has been studied broadly during the COVID-19 pandemic, and it has been shown to arise from an unattainable ideal of motherhood that mothers compare themselves to (Constantinou et al. 2021). Our study also indicates that maternal guilt deepened and intensified because changes in living conditions drained mothers' resources, as supported by previous research (Thorsteinsen et al. 2022).

### 5.2. Health Concerns

All respondents stated that their health concerns increased due to the COVID-19 pandemic. Contracting the virus was described as one of the biggest concerns, targeted at immediate family members but also more widely at vulnerable groups and society:

> *I worry about the mental health of my children. I'm afraid my children will be infected and suffer. What happens if one of us parents die (we have no support here and all scenarios seem impossible). What happens if one of us parents gets infected and is sent to the hospital? What if our relatives abroad die? What will happen to society? How will vulnerable groups be affected?* (16)

As this example illustrates, health concerns were related to mental and physical health, the consequences of restriction measures, as well as mothers' responsibilities as caregivers. Many respondents brought up a worry over who would take care of their children if they themselves got sick because they lacked social contacts and support in Finland. This concern was visible in both mothers who lived with and without a partner.

Many respondents elaborated that the consequences of the lockdown caused stress and anxiety alongside additional care and domestic work and supporting their children.

> *Everyone's stress levels are high, all children show signs of anxiety. Two of the children present consistent regression behavior. We parents also must cope with higher stress, partly due to the changes the situation has brought, as well as to the general anxiety, having to cope with a huge load of responsibilities and having to support the children emotionally.* (16)

The participants linked their feelings of stress and anxiety to the fear of contracting COVID-19, the need to figure out and decide how to best protect their children and themselves from the virus, and to the uncertainty of the duration and consequences of the pandemic. Social isolation deepened these concerns because it was not possible to explore emotions face-to-face with anyone other than immediate family members.

The responses point out that being foreign speaking caused uncertainty over whether the mothers were receiving the necessary information about national public health instructions and restrictions (see also Backholm and Nordberg 2023). This further intensified health concerns, as the following quote illustrates:

> *Our Finnish is only mediocre and so we always felt like we didn't get all the information.* (28)

Some of the respondents interpreted not receiving understandable information as being left alone with their concerns without any support, or information about support, available to them. This increased stress over their parental care responsibilities, as it is against the moral expectation that parents should be capable of finding and obtaining the right services and information concerning their children's health, well-being and security (see also Heino et al. 2023a).

*5.3. Increased Care and Domestic Work*

In addition to all daily activities taking place in one physical space, these also happened simultaneously. The example below vividly shows how boundaries between different spheres of life became blurred:

> *In one moment, I just realized that all aspects of my everyday life have merged together.* (51)

These descriptions revealed how exhausting it is to simultaneously handle one's work and/or study and take care of children and other household duties, such as cleaning and cooking. These difficulties concerned most participants, but especially those who had more than one child because it was difficult to divide their time between children. Single mothers brought up the lack of another person to share the increased responsibilities with, which caused constant time pressure. While care and domestic workloads increased, it was not possible to outsource domestic labor because of restrictions.

The increase in household and childcare tasks was described, for example, by an increased need to cook. This required more time than before since all family members ate their meals at home. Meals require planning, shopping, preparation, and cleaning afterwards. Respondents generally described themselves as taking on an additional burden for performing domestic work tasks, as mothers, and acting as the main organizers of all domestic and care work:

> *As a mother I must make sure that the youngest child gets his homework done. I need to control that all things get done.* (65)

Some of the mothers described the pursuit of more egalitarian arrangements after the first weeks of the pandemic. While this might have sparked some changes, the respondents continued to carry the main responsibility for domestic and care work:

> *In the beginning we had conflicts, for I (mom) had to take care of our child's studies and other household work (dishwashing, laundry etc.) Now my husband tries to help more by buying foods and doing exercise with our child.* (14)

As the extract above shows, a reorganization of the workload did not necessarily mean the equal distribution of responsibilities and duties between spouses, since the respondent describes how her husband "tries to help" with some tasks. In some answers, a greater equality in task-sharing was visible from the use of "we" when discussing the additional burden:

> *We take turns looking after and caring for our 3-year-old child, and we don't have free time for ourselves anymore, because when we are not taking care of the child, we have to work.* (15)

Mothers described assisting their children with studying remotely as "*difficult*" or "*impossible*". This was often because of insufficient skills in the language of the schoolwork. Mothers also wrote about the difficulties of combining their own remote work and the additional burden of domestic and care work.

> *Parents must do a lot of things quickly and immediately, because the child is not able to do many things on his own, to be responsible for learning during quarantine. Cognitive distortion occurs when requirements are high but resources and means to fulfill the requirements are not enough in reality.* (71)

Overall, this new situation with additional care and domestic work and constant interruptions from family members caused pressure for mothers to multitask and repeatedly switch between tasks, which, according to them, sparked feelings of dissatisfaction in all areas of their everyday lives:

> *I feel like I'm not performing at my best level in school, work or even mommy job. I feel "sooo" torn and exhausted.* (34)

Our results are in line with other studies (Craig and Churchill 2021; Hjálmsdóttir and Bjarnadóttir 2020) showing that mothers reported an increase in unpaid work and

multitasking during the COVID-19 pandemic. This led to conflict between their roles as a mother and a worker and blurred the boundaries between different spheres of life, leading to emotional exhaustion. Assisting children with remote schooling was difficult when there was a lack of skills in the language of the schoolwork. Our analysis also revealed a worry over what might happen to the children should the mothers themselves contract the virus because of the lack of support networks in Finland.

## 6. Loving

### 6.1. Tense Relationships with Other Household Members

Most respondents stated that the COVID-19 pandemic strained their relationship with their children and spouses because they lacked resources and time and were "*constantly irritated*", "*tired*" and "*stressed*", which was visible in communication. As one mother describes:

> *"I simply do not have time for relationships, all my time goes into surviving".* (63)

When in "survival mode", one only performs essential domestic and work tasks and yet constantly feels tired. This was reflected in family relations, and, according to the respondents, was an experience other family members shared.

> *My husband and I do argue a little more over smaller things, but we are trying to understand and recognize that this is simply because we are around each other all the time. The eldest child is not happy about being home all the time, and some of the outdoor activities she is used to are closed, so it is hard for her to adjust. That has caused some outbursts and some issues which have then caused issues between my partner and me.* (33)

Increased conflicts between the mothers and their children or partners were explained by altered routines, excessive time spent together, the stress that the mothers were experiencing due to multitasking, the unpredictability of the pandemic, as well as their children's behavior, such as increased tantrums.

> *My child is throwing tantrums every day because of the change in routines, and because he is bored. We try to keep him entertained but he misses playing with kids of his age and going to kindergarten. The number of outbursts is affecting seriously my mental health, to the point that some days, at the end of the day, I don't want to be anywhere near my child.* (15)

In the above example, a mother describes conflicting emotions, as she shows empathy for the child but also acknowledges that the child's behavior negatively affects her own mental health. Overall, the pandemic forced families to spend more time together. Family members also shared this experience of changed routines. This, in some cases, brought them closer, which means that the pandemic had both negative and positive effects on family relations.

> *There are highs and lows. We are getting tired of each other but also bonding a lot.* (32)

In the accounts, close family relations seemed to require different arrangements between spouses, like providing privacy for all family members. These actions gave the mothers the resources to read the emotions of others and regulate their own emotions, which reduced conflicts. Our results illustrate the effects that family relations have on mothers' well-being, as they may cause strain but also provide a sense of connection as well as emotional and practical support.

### 6.2. Disrupted Face-To-Face Relationships with Family Living Abroad

Mothers described that their relationships with relatives living abroad, especially older relatives, were filled with worries because restrictions changed the travelling practices of many families:

> *What I am most worried about is the health of my ageing parents in my home country and that I cannot travel to be with them if my help is needed.* (67)

The respondents described that, in the case of illness in the family, carrying out their care responsibilities would require physical presence, which was impossible because Finland's borders were closed. Caring for their aged parents in other countries was described as a moral and practical responsibility. This involved the expectation of organizing and providing physical care as well as emotional support when needed:

> *I feel anxiety for my home country. I know it will be a long time until I can see my family and friends in my home country. It is heartbreaking to see it from a distance.* (31)

Travelling to their country of origin was described as an opportunity to nurture certain cultural practices within the family and maintain family ties, which was no longer possible:

> *To us travelling abroad is not just vacation like for tourists, it is very important. It is about going home for a little while. It is about introducing our children to their relatives and their culture/language/customs.* (25)

According to the respondents, an emotional bond to one's country of origin is linked to memories, family ties, and the possibility to freely speak one's mother tongue and practice different cultural customs; the respondents also wanted to provide their children with this. Technology and remote communication have facilitated staying in touch, but as one respondent described, "*to be truly connected requires face-to-face presence*". In the data, feelings of homesickness, cultural mourning, and worries related to the pandemic's development and the health of family members were often connected with anxiety.

### 6.3. Weakening Ties to Finnish Society

Mothers stated that isolation had been a part of their lives as migrants even before the pandemic. According to them, this is especially the case if a migrant does not speak Finnish fluently or have social contacts in Finland. Mothers stated that the pandemic had deepened this isolation and the feeling of being outside of society, especially for mothers who were on maternity leave or only had social contact through work, studies, and social and health services, or who had only recently moved to Finland:

> *I feel like we are entirely disconnected from Finland, we don't see so many Finnish people in our daily life, the school and daycare were a big part of our lives, and it is very different now when they are closed.* (31)

Social isolation created an experience of being disconnected from Finnish society, even though the respondents physically resided in Finland. Everyday life without social connections to society was described as "*empty and meaningless*". Mothers were also worried about the effects this isolation would have on their children:

> *As foreigners in Finland, they need to stay in touch with their classmates, for the language but also for social connections. We are already 'different', and being in our own bubble will only increase this difference and isolation. This has affected the mental health of us all.* (4)

Another concern was the effect on their own Finnish language skills, and language was described as an important means of becoming part of Finnish society:

> *I don't have the possibility to go to Finnish language classes. My language skills are declining, I am not speaking Finnish with anyone, and soon I will forget it. This worries me a lot. The support for migrant families in this sphere is very weak.* (67)

The described sense of isolation from Finnish society produced feelings of loneliness. This was exacerbated by the fact that mothers were not offered any official support to maintain a connection to Finnish society and practice the Finnish language. Isolation was seen as a cause of stress because it concerned not only the mothers but also their children, and was seen as something difficult to have control over.

## 7. Being

### 7.1. Lack of Alone and Free Time

Most of the respondents reported a lack of time for themselves and being alone because daily tasks were planned around the needs of the children, as well as around work and domestic tasks. The increased burden of domestic and care tasks led to constant time pressure:

*I lack free time. Whenever I have free time (meaning that my partner is taking care of our child) I must work, study, cook, go shopping, and do the house chores. I do not have time to relax or do things I enjoy.* (15)

Mothers stated that a lack of free time meant a decrease in their quality of life since there was no time for things that brought joy before the pandemic, such as reading, watching movies, doing sports, or studying. As one mother describes it:

*The pandemic has changed our life because we don't feel completely free.* (46)

Not being 'free' was brought up by several respondents. It was connected to not being able to do what one wanted because one had to constantly serve others, that is, children, employers and partners. A lack of own and free time was described as one of the major changes brought by the pandemic. With no free time, it was not possible to "*relieve stress*", "*relax*" and "*gain strength*".

### 7.2. Family-Centered Activities

Many mothers described efforts to make the best of the situation:

*We could spend quality time with each other. We evolved as one family, which was not possible before because of the very routine with schools and offices.* (13)

The pandemic also brought positive changes; it cleared up busy schedules since family members no longer commuted to work or hobbies, and demands in that sense decreased. The pandemic also provoked people to think about the fragility of life, their priorities and ways of life because COVID-19 was life threatening:

*The whole family just stopped unexpectedly and realized that we did not live in the right way before. We all were just running and driving somewhere all the time!* (80)

Respondents stated that being present for each other provided them emotional support in many ways: it brought joy and relieved stress in a difficult situation. Spending time with the family included talking, playing games, watching movies, cooking, and doing sports and outdoor activities:

*We spend more time talking and more time together. We workout together and put more effort to keep calm and support each other to overcome "boredom" at home.* (20)

However, spending relaxed time with other family members required the mothers to plan, schedule and coordinate their own and the whole family's activities:

*I try to have a time-schedule for studies, doing assignments, cooking, playing with my child, talking with the family.* (2)

In all the responses, the mothers described themselves as active in initiating and up-holding family-centered activities and guiding children in their daily tasks. In other words, they were performing cognitive labor, which included anticipating the need for actions in the family's everyday life, implementing those actions and monitoring the progress (Daminger 2019). Our results show that cognitive labor included the practical element of implementing different responsibilities from planning to actions and also emotional labor, such as managing their children's emotions.

## 8. Discussion

In this study, we have explored the changes that the COVID-19 pandemic brought to foreign-speaking migrant mothers' well-being. Our results show that the initial stage of the pandemic affected migrant mothers' well-being and mental health in multiple ways.

Allardt's (1976) classical framework was useful for the analysis of our data. He conceptualized well-being as multidimensional, requiring decent living conditions, social bonds and fulfillment of the self. In our analysis, we have highlighted how the pandemic generated ill-being along intersectional divides for our research participants. The decreasing sense of well-being emerged as deteriorated daily living conditions; health concerns; increased responsibility for unpaid care and domestic work; a mismatch between expectations concerning, for example, support in home schooling and following official information on restrictions and mothers' abilities to fulfill these expectations due to language, unsettled local and transnational family relationships; weakening ties to Finnish society; and a lack of one's own and free time. However, there were also accounts of positive impacts on well-being in everyday life, for example, related to increased family time.

Other scholars have expanded Allardt's framework with new dimensions, for example, with 'doing' (Helne and Hirvilammi 2022) and 'interaction' (Moses 2022), both of which are related to how individuals act in their environment. This study contributes to the previous literature by recognizing the important notions of autonomy and control for the well-being of structurally vulnerable people in states of exception or crisis. The analysis shows how, along with the participants' daily living conditions deteriorating, public support from authorities, such as social workers, became less accessible (see also Harrikari et al. 2023). Moreover, the situation of isolation or loneliness followed by this was further exacerbated by language barriers. To satisfy one's needs and the quest for autonomy in a situation with diminishing public support, informal social networks and support, 'loving', become more strongly interconnected with the dimensions of 'having' and 'being'. For the participating mothers, social support was often the structurally weakest link, and a lack of local social relations and support outside family added to their anxiety related to the security and health of their family.

Our results align with previous studies concerning mothers in the majority population in many aspects (e.g., Wandschneider et al. 2022; Thorsteinsen et al. 2022). The pandemic seems to have reinforced previous gender inequalities (Czymara et al. 2021). However, our results also highlight the complex intersectional nature of everyday life in a pandemic, showing how various linguistic disadvantages and migrant backgrounds structured the well-being of the mothers, even though most of them were relatively highly educated. In the context of the pandemic, migrancy was firstly linked to language barriers. For example, non-Finnish or Swedish-speaking mothers experienced difficulties accessing accurate health and public authority information and helping their children with distance learning, leading to a situation of educational inequality for foreign-speaking children. Secondly, migrancy was associated with a hyphenated sense of vulnerability related to a lack of social networks and support—a sense of isolation and homesickness that brought with it cultural mourning and worries about the health of family living in other countries.

It is noteworthy that the growing responsibilities remained a private matter for the mothers to deal with on their own. When facing difficulties in handling all (impossible) demands, many mothers turned inwards, experiencing maternal guilt, stress, inadequacy, and anxiety, which affected their mental health. We align with the view of Ghosh and Chaudhuri (2023), who highlight that care should be a social issue, rather than a private matter, that focuses on the structures that create uneven distributions of care and domestic work, demanding changes on a structural level. In addition, minority women and mothers are rarely at the center of discussion regarding policy planning. An intersectional perspective incorporating divisions of gender, race, social class and ability should be included in those discussions.

From a practical perspective, it is essential that foreign-speaking mothers are provided with information about the care alternatives available to them should they become sick

and unable to care for their children. Information about public decisions, restrictions and recommendations should be accessible in different languages. In future pandemics, it is crucial to provide additional support to foreign-speaking mothers during remote education and find ways to reduce the isolation of families and provide emotional support when needed.

### 9. Limitations and Future Developments

This research has some limitations. By using an electronic survey, we aimed to allow the participants to respond without disrupting their daily lives. The purpose of open-ended questions was to encourage the participants to reflect on their experiences and bring up issues relevant to them. However, this form of data production also requires accessible internet and electronic devices, as well as the ability, willingness and time to write longer answers. Consequently, the data potentially excluded mothers who did not have these resources. The questionnaire was also limited to particular languages, excluding large migration population groups.

The most important question at this moment and in the future asks whether the negative changes in migrant mothers' well-being shown in this and other studies were temporal or whether there will be long-term consequences in mothers' everyday lives. Finally, future research should also pay explicit attention to the experiences of migrant children during and after the COVID-19 pandemic.

**Author Contributions:** Data collection, E.H. and H.K.; conceptualization, E.H. and H.K.; methodology, E.H. and H.K.; analysis, E.H., H.K. and C.N.; writing—original draft preparation, E.H., H.K. and C.N.; writing—review and editing, E.H., H.K. and C.N. All authors have read and agreed to the published version of the manuscript.

**Funding:** The authors received financial support for the research from the Ministry of Social Affairs and Health, Finland, project number VN/13883/2021, and Academy of Finland project number 310610 and 334686. Open access funding provided by the University of Helsinki.

**Institutional Review Board Statement:** The study was conducted according to the guidelines of the Declaration of Helsinki and followed guidelines on research ethics ascribed by Finnish Academy and The Finnish National Board on Research Integrity TENK. Ethical review and approval were waived for this study, because the study does not contain any of the elements that require the ethical review by the guidelines of the Åbo Akademi University. Ethical review in our system is required only in ethically challenging research designs, for example if it involves minors, participants are exposed to exceptionally strong stimuli, research poses security or risk to mental harm for participants beyond the risks encountered in normal life.

**Informed Consent Statement:** Informed consent was obtained from all subjects involved in the study.

**Data Availability Statement:** The participants of this study did not give consent for their data to be shared publicly, so due to the sensitive nature of the research supporting data is not available.

**Conflicts of Interest:** The authors declared no potential conflicts of interest with respect to the research, authorship, and/or publication of this article.

### Appendix A Everyday Life of Families with Children during the Coronavirus Lockdown
### Open questions:

(1)    How has the Coronavirus lockdown affected your family's everyday life?
(2)    How has the lockdown affected your relationship with your partner and/or your children?
(3)    What kind of strain has the lockdown brought to your family's everyday life?
(4)    What kind of concerns or worries have been most prominent for you during the lockdown?
(5)    What kind of coping strategies have you developed in this situation?
(6)    Have you felt in need of external support or assistance in this situation? What type of support or assistance?
(7)    Have you received assistance during lockdown? What type of assistance and from where?
(8)    Have you assisted others in this situation? Who have you assisted and in what way?

(9) Has the lockdown had any positive effects on your family's everyday life? In what ways?

**Background questions:**

(1) Age.
(2) Level of education.
(3) How many adults live in your household?
(4) How many children live in your household?
(5) How old are the children? (You may choose several options according to your situation).
(6) Region.
(7) Country of birth.

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
