# Peer review of "Changes in the Well-Being of Foreign Language Speaking Migrant Mothers Living in Finland during the Initial Stage of the COVID-19 Pandemic"

_socsci, doi:10.3390/socsci13010042_

Round 1
Reviewer 1 Report
Comments and Suggestions for Authors
Firstly, thank you for giving me the opportunity to review this paper.
The article "Changes in the well-being of foreign language speaking mothers living in Finland during the initial stage of the COVID-19 pandemic" deals with a topic that is appropriate for the journal Social Sciences. However, specific references to migration, mental health and differences between men and women and between natives and immigrants should be included, given the aim of the study.
Regarding the results, many of them relate to situations that most mothers have experienced, regardless of whether they are migrant or not. If the research is carried out with a migrant population and the aim is to show the greater difficulties they have compared to non-migrant women or compared to men (migrant or not), it would be necessary to put more emphasis on what is specific to migrant women.
In the discussion, I would differentiate between what can be generalised to the situation of women and/or mothers and what is specific to the migrant population, because there is an intersection between woman, mother, migrant, etc. This would provide tools for understanding the process and for intervention. However, this would require a section in the introduction explaining the relationship between migration and well-being or mental health, and between the differences between men and women in the migration process and in mental health (if possible, work done in Finland or in Europe). This would better contextualise the work. Without an emphasis on the migration part, the aim of the study may be diluted.
The methodology would also need to be more specific.
Suggestions include the following:
1. Abstract: It is not clear until the end that the study population is migrant. I am sure there is a reason to talk about mothers who speak a foreign language, but I suggest to make it clearer about migrations for bibliographic searches.
2. Keywords: add migration or migrants.
3. Introduction: The section on COVID and health consequences, especially for women/mothers, is good. But I miss a section on migration, consequences, differences between men and women, and well-being or mental health.
In this sense, I would talk about culture shock or the stress of migration (all the work of John Berry, or Ward and Kennedy, 1992, 1999, for example). I would also have in mind the grief of migration (some of the work of Achotegui, for example). And then, the differences in the process and health between men and women migrants, and between natives and migrants (for example in Europe: Aroian, Norris, González de Chávez and Garcia, 2008; Elgorriaga, Ibabe and Arnoso, 2019; Jasinskaja-Lahti, Liebkind and Perhoniemi, 2006; Singhammer and Bancila, 2011; Singh et al., 2015).
6. Procedure: It would be appropriate to further specify the procedure followed for collecting the information and for the analysis, as it is important to understand the results. Some questions that arise
a. What are the 9 questions that were asked, were they directed to look for the three dimensions or were they open-ended?
b. How many people were involved in analysing the answers and was there agreement between the judges?
4. Results: I would put the results in another section and perhaps make a diagram or table with the three dimensions and sub-sections to make it clearer.
And as I said before, many of the findings can be generalised to other mothers, sometimes women, and in some cases to anyone. For this reason, more emphasis should be placed on the specificities of migration and the intersection of different situations of vulnerability.
5. Discussion: include the points made in the previous sections.
Author Response
REVISION AND RESUBMISSION OF MANUSCRIPT
We thank you for the opportunity to revise and resubmit our manuscript. The suggestions offered by the reviewers have proved helpful and resulted in an improved manuscript. We appreciate the time and effort that the reviewers dedicated to providing their constructive feedback.
In what follows, we have included the reviewers’ comments and our point-by-point responses to their comments (our responses appear in italics). In these responses, we have indicated how we have addressed each concern, and we describe the changes we have made to the manuscript. The revisions to the manuscript have been approved by all the co-authors. In the manuscript changes are marked in red font.
We hope that this revised version meets with your approval, and we look forward to hearing from you regarding our resubmission. We thank you for considering our manuscript for publication once again and would be happy to respond to any further questions or comments you may have.
Warmest regards,
The authors
Reviewer 1
- The article "Changes in the well-being of foreign language speaking mothers living in Finland during the initial stage of the COVID-19 pandemic" deals with a topic that is appropriate for the journal Social Sciences. However, specific references to migration, mental health and differences between men and women and between natives and immigrants should be included, given the aim of the study.
Thank you for bringing this up. We added this aspect to the Introduction, and we added a section after the Introduction called “Migrants’ well-being during COVID-19". We have referred to the research participants as foreign speaking, since their position as non-dominant language speakers and as mothers was the focus of our study. However, we have tried to clarify this by discussing our focus in the introduction, and then throughout the paper referring to the research participants as “migrant mothers”.
- Regarding the results, many of them relate to situations that most mothers have experienced, regardless of whether they are migrant or not. If the research is carried out with a migrant population and the aim is to show the greater difficulties they have compared to non-migrant women or compared to men (migrant or not), it would be necessary to put more emphasis on what is specific to migrant women.
This is true. We added these perspectives in introduction, analysis chapter and to the discussion. We also added table on p. 6 in which we clarify language and migration related issues. While our results indicate that this specific group of foreign speaking migrant mothers also had a lot in common with mothers of the majority population.
- In the discussion, I would differentiate between what can be generalised to the situation of women and/or mothers and what is specific to the migrant population, because there is an intersection between woman, mother, migrant, etc. This would provide tools for understanding the process and for intervention. However, this would require a section in the introduction explaining the relationship between migration and well-being or mental health, and between the differences between men and women in the migration process and in mental health (if possible, work done in Finland or in Europe). This would better contextualise the work. Without an emphasis on the migration part, the aim of the study may be diluted.
Thank you for this suggestion, we added this section in the introduction. We also added section after introduction called: “Migrants’ well-being during COVID-19”.
- The methodology would also need to be more specific.
We acknowledged this comment in the methodology section.
- Abstract: It is not clear until the end that the study population is migrant. I am sure there is a reason to talk about mothers who speak a foreign language, but I suggest to make it clearer about migrations for bibliographic searches.
Thank you for this suggestion. We have clarified this and included a discussion on migrants and migration earlier in the paper.
- Keywords: add migration or migrants.
We made this addition.
- Introduction: The section on COVID and health consequences, especially for women/mothers, is good. But I miss a section on migration, consequences, differences between men and women, and well-being or mental health.
This is a great suggestion. We added a new section after Introduction called “Migrants’ well-being during COVID-19”.
- In this sense, I would talk about culture shock or the stress of migration (all the work of John Berry, or Ward and Kennedy, 1992, 1999, for example). I would also have in mind the grief of migration (some of the work of Achotegui, for example). And then, the differences in the process and health between men and women migrants, and between natives and migrants (for example in Europe: Aroian, Norris, González de Chávez and Garcia, 2008; Elgorriaga, Ibabe and Arnoso, 2019; Jasinskaja-Lahti, Liebkind and Perhoniemi, 2006; Singhammer and Bancila, 2011; Singh et al., 2015).
Thank you for this suggestion. We do understand what you mean. However, our field of discipline is social policy and social work and even though we are speaking about well-being, we try to focus on a more structural approach, and as we now state in the Introduction, we understand migration as structural phenomena, meaning that social structures can create or uphold different form of vulnerabilities of the migrant population. We have clarified this in the Introduction.
- Procedure: It would be appropriate to further specify the procedure followed for collecting the information and for the analysis, as it is important to understand the results. Some questions that arise a. What are the 9 questions that were asked, were they directed to look for the three dimensions or were they open-ended? b. How many people were involved in analysing the answers and was there agreement between the judges?
This is a very good point. We added the survey questions as an appendix. We also explained our analysis and the task division between the authors as follows: “We aimed to create a dialogue between our observations from the empirical data and well-being theory of Allardt. After reading the data, we continued our analysis through several steps. The first author created an initial coding chart which included the classification of the data by three dimensions of well-being. Next, the first and second authors examined and worked on this division, and tackled the similarities and differences they recognized, creating a revised coding chart aimed at more explicitly addressing and naming themes within each well-being dimension. In the final stage, all three authors reflected and elaborated each theme within the well-being framework. At this stage we especially focused on the changes the participants described in their well-being, since change was central to the responses.”
- Results: I would put the results in another section and perhaps make a diagram or table with the three dimensions and sub-sections to make it clearer.
Thank you for the excellent suggestion. We added a table of the results on page 6.
- And as I said before, many of the findings can be generalised to other mothers, sometimes women, and in some cases to anyone. For this reason, more emphasis should be placed on the specificities of migration and the intersection of different situations of vulnerability.
We agree and rewrote the discussion accordingly.
- Discussion: include the points made in the previous sections.
Noted. Thank you for your time, support and great suggestions!
Reviewer 2 Report
Comments and Suggestions for Authors
The project reported in this paper is interesting and I am sure it will make an important contribution to research on migrant women. However, revisions need to be made to ensure clarity and robust critical analysis. Please see below for some feedback along these lines:
The language and phrasing used by the author(s) tends to homogenize the diverse population of migrant women. The assumption that all migrant women experience barriers and oppression in the same, homogeneous way is quite problematic. Careful rephrasing is needed, as well as more careful use of intersectional approaches, with minimal critical analysis of how the diverse backgrounds have shaped (or are shaped by) their migration experiences is problematic. This should also be reinforced in the bibliography.
well-being model used as a theoretical framework needs to be more complex, as there are many studies on this topic that have made significant contributions to the field of migration, including in the context of COVID. This is particularly important in the discussion. Some examples that may help the authors are:
1. Garrido, R., Paloma, V., Benítez, I., Skovdal, M, Verelst, A. & Derluyn, I. (2023): Impact of COVID-19 pandemic on the psychological well-being of migrants and refugees settled in Spain, Ethnicity & Health, https://doi.org/10.1080/13557858.2022.2035692
2. Wang, F., Tian, C., & Qin, W. (2020). The impact of epidemic infectious diseases on the wellbeing of migrant workers: a systematic review. International Journal of Wellbeing, 10(3).
In terms of methodology, it is recommended that the full qualitative survey be included in the article as an appendix.
Additionally, a relational graph of the results that deepens the relationships between the contents would help to deepen the qualitative analyses.
The discussion is weak and needs to be strengthened considerably, especially in the following directions. First, more explanation of the results is needed, providing a better contextualization of how COVID-19 affected the country and the measures that were implemented, and how this did or did not reach the migrant population. Secondly, an intersectional analysis should be included in the discussion of the results, in terms of cultural or ethnic origin, socio-economic status, migratory status... without this, talking about women sounds naive and raises more doubts than answers. The work of Hill-Collins can help broaden the vision. Third, it is necessary to include a deep reflection on the limitations of the study, which will lead us to understand its gaps and to think about future lines of research.
Finally, it is recommended to check the formatting as there are some errors (e.g., quotation marks with double spaces, page 11, line 524, a hyphen is missing).
Author Response
REVISION AND RESUBMISSION OF MANUSCRIPT
We thank you for the opportunity to revise and resubmit our manuscript. The suggestions offered by the reviewers have proved helpful and resulted in an improved manuscript. We appreciate the time and effort that the reviewers dedicated to providing their constructive feedback.
In what follows, we have included the reviewers’ comments and our point-by-point responses to their comments (our responses appear in italics). In these responses, we have indicated how we have addressed each concern, and we describe the changes we have made to the manuscript. The revisions to the manuscript have been approved by all the co-authors. In the manuscript changes are marked in red font.
We hope that this revised version meets with your approval, and we look forward to hearing from you regarding our resubmission. We thank you for considering our manuscript for publication once again and would be happy to respond to any further questions or comments you may have.
Warmest regards,
The authors
Reviewer 2
- The language and phrasing used by the author(s) tends to homogenize the diverse population of migrant women. The assumption that all migrant women experience barriers and oppression in the same, homogeneous way is quite problematic. Careful rephrasing is needed, as well as more careful use of intersectional approaches, with minimal critical analysis of how the diverse backgrounds have shaped (or are shaped by) their migration experiences is problematic. This should also be reinforced in the bibliography.
Thank you for pointing this out. This is of course a concern that we take seriously. We wrote a paragraph in the introduction in which we explain how we approach migration. We also tried to paraphrase some of the expressions that can be perceived as othering, such as they in comparison to us.
- well-being model used as a theoretical framework needs to be more complex, as there are many studies on this topic that have made significant contributions to the field of migration, including in the context of COVID. This is particularly important in the discussion. Some examples that may help the authors are:
- Garrido, R., Paloma, V., Benítez, I., Skovdal, M, Verelst, A. & Derluyn, I. (2023): Impact of COVID-19 pandemic on the psychological well-being of migrants and refugees settled in Spain, Ethnicity & Health, https://doi.org/10.1080/13557858.2022.2035692
- Wang, F., Tian, C., & Qin, W. (2020). The impact of epidemic infectious diseases on the wellbeing of migrant workers: a systematic review. International Journal of Wellbeing, 10(3).
Thank you for this suggestion. After Introduction we added section called “Migrants’ well-being during COVID-19” where we discuss this issue.
- In terms of methodology, it is recommended that the full qualitative survey be included in the article as an appendix.
Thank you for this recommendation. All questions are included as an appendix after the references.
- Additionally, a relational graph of the results that deepens the relationships between the contents would help to deepen the qualitative analyses.
We added a table of the results at page 6.
- The discussion is weak and needs to be strengthened considerably, especially in the following directions. First, more explanation of the results is needed, providing a better contextualization of how COVID-19 affected the country and the measures that were implemented, and how this did or did not reach the migrant population.
This is a valid comment, we have included more information about how COVID-19 affected Finland, and what measures were taken in relation to the migrant population. We also included a more thorough discussion on the findings.
- Secondly, an intersectional analysis should be included in the discussion of the results, in terms of cultural or ethnic origin, socio-economic status, migratory status... without this, talking about women sounds naive and raises more doubts than answers. The work of Hill-Collins can help broaden the vision.
Thank you for this suggestion. We rewrote all sections from this perspective, especially discussion. However, we did not add intersectional approach as a method or tool explicitly at this point. This is because of limitations of our data to do thought intersectional analysis.
- Third, it is necessary to include a deep reflection on the limitations of the study, which will lead us to understand its gaps and to think about future lines of research.
We added an additional section on limitations and future directions after the discussion.
- Finally, it is recommended to check the formatting as there are some errors (e.g., quotation marks with double spaces, page 11, line 524, a hyphen is missing).
We made these changes to the text.
Thankyou for good suggestions. We sincerely appreciate all your comments
Round 2
Reviewer 1 Report
Comments and Suggestions for Authors
congratulations
Reviewer 2 Report
Comments and Suggestions for Authors
The issues raised in the review have been effectively addressed by the authors.